

# Thermalization and hydrodynamics of two-dimensional quantum field theories

**Luca V. Delacrétaz[1], A. Liam Fitzpatrick[2], Emanuel Katz[2] and Matthew T. Walters[3]**

**1** Kadanoff Center for Theoretical Physics, University of Chicago,
Chicago, IL 60637, USA
**2** Department of Physics, Boston University,
Boston, MA 02215, USA
**3** Institute of Physics, École Polytechnique Fédérale de Lausanne (EPFL),
CH-1015 Lausanne, Switzerland

## Abstract

We consider 2d QFTs as relevant deformations of CFTs in the thermodynamic limit. Using causality and KPZ universality, we place a lower bound on the timescale characterizing the onset of hydrodynamics. The bound is determined parametrically in terms of the temperature and the scale associated with the relevant deformation. This bound is typically much stronger than $\frac{1}{T}$, the expected quantum equilibration time. Subluminality of sound further allows us to define a thermodynamic $C$-function, and constrain the sign of the $\mathcal{T}\bar{\mathcal{T}}$ term in EFTs.

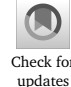
# 1   Introduction and Bound on Thermalization

We consider relativistic quantum field theories (QFTs) in two spacetime dimensions obtained from UV conformal field theories (CFTs) with a relevant deformation

$$\lambda \int d^2x\, \mathcal{O}\,, \qquad \Delta \leq 2\,. \tag{1}$$

In the high temperature limit, the equation of state of the QFT approaches that of the UV CFT, so that in particular the speed of sound approaches the conformal value $\lim_{T\to\infty} c_s = 1$. Causality leaves no room for hydrodynamic spreading about the sound front when $c_s = 1$, and one finds indeed that thermal correlation functions of CFTs in the thermodynamic limit are entirely fixed by symmetry [1]. Instead, at intermediate temperatures, the QFT is expected to enter a hydrodynamic regime at late times $t \gg \tau_{\rm eq}$ with speed of sound $c_s < 1$; however the near-luminal speed of sound will place important constraints on the equilibration time $\tau_{\rm eq}$ at which hydrodynamics emerges.

In Ref. [2], Hartman, Hartnoll and Mahajan showed – following earlier work on hydrodynamics and causality [3–5] – that the vanishing of commutators outside the lightcone $x > ct$ imposes a parametric bound on $\tau_{\rm eq}$ in diffusive systems. The bound arises because diffusion $x^2 \sim Dt$ is superluminal at early times, and hence cannot emerge too soon. It reads

$$\tau_{\rm eq} \gtrsim \frac{D}{c^2}\,, \tag{2}$$

where $D$ is the diffusion constant. A simple generalization of this bound for hydrodynamic modes with dispersion relation $\omega \simeq c_s k - i\mathcal{D}k^z$ (instead of $\omega \sim -iDk^2$) is

$$\tau_{\rm eq} \gtrsim \frac{\mathcal{D}^{1/(z-1)}}{(c-c_s)^{z/(z-1)}}\,. \tag{3}$$

In the following we set $c = 1$. As we review below, the hydrodynamics of interacting QFTs in two spacetime dimensions is governed by the KPZ universality class, with $z = 3/2$ rather than $z = 2$ for diffusion (note that $z$ is unrelated to the zero temperature dynamic critical exponent, which for relativistic QFTs is always unity). Eq. (3) captures the absence of hydrodynamics in 2d CFTs, since it requires $\tau_{\rm eq} \to \infty$ when $c_s \to 1$. For QFTs, corrections to the equation of state from breaking of conformality at intermediate temperatures will control $1-c_s$. They also control $\mathcal{D}$, as dissipation in the KPZ universality class is entirely fixed by thermodynamics. Putting these together, one finds that at large but finite temperature $T \gg \lambda^{1/(2-\Delta)} \equiv \Lambda$, thermalization is allowed but strongly constrained

$$\tau_{\rm eq} \gtrsim \frac{1}{T}\frac{1}{c_{\rm UV}}\left(\frac{T}{\Lambda}\right)^{2(2-\Delta)}\,, \tag{4}$$

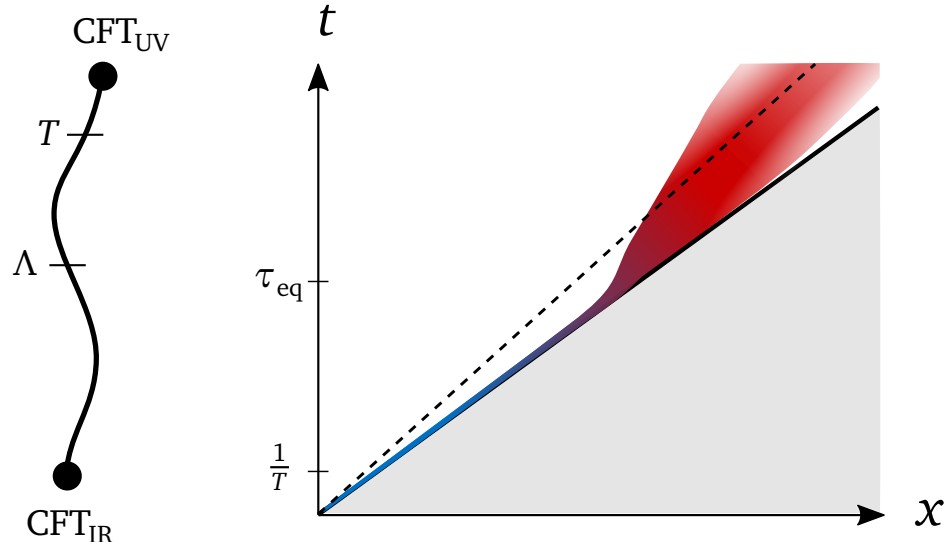

Figure 1: *Left:* 2d QFT obtained from a UV CFT and a relevant deformation with scale $\Lambda$. *Right:* Sketch of a correlator (e.g., $\langle \mathcal{T}_{00}(t,x)\mathcal{T}_{00}\rangle$) at high temperatures $T \gg \Lambda$. The solid line is the lightcone $x = t$. The correlator is well described by the CFT expression at early times $t \lesssim \tau_{\text{eq}}$, and by KPZ hydrodynamics at late times $t \gtrsim \tau_{\text{eq}}$ where a sound front emerges along the dashed line $x = c_s t$.

where $c_{\text{UV}}$ is the central charge of the UV CFT. We argue that this bound is in fact parametrically saturated, as long as $c_{\text{UV}} = O(1)$ and the RG flow is not near-integrable. Similar arguments show that thermalization is also strongly constrained at low temperatures. At intermediate temperatures, the bound depends on the details of the equation of state which can no longer be accessed with conformal perturbation theory, but generically requires

$$\tau_{\text{eq}} \gtrsim \frac{1}{T}\frac{1}{s_o}\left(\frac{d\log s_o}{d\log T}\right)^{-1}, \tag{5}$$

where $s_o(T) \equiv s(T)/T$ is the dimensionless entropy density. The last factor is sensitive to conformal breaking: it is typically $O(1)$ at intermediate temperatures $T \sim \Lambda$, but becomes large at high temperatures (where one recovers Eq. (4)) and low temperatures. The precise expression for this thermodynamic factor is given in Eq. (42).

It has been conjectured that the equilibration times of interacting quantum systems satisfy a quantum bound $\tau_{\text{eq}} \gtrsim \frac{1}{T}$ [6] – our results show that all 2d QFTs that do not have a large number of degrees of freedom satisfy this bound, and satisfy a parametrically stronger bound (4) at high or low temperatures (for marginally relevant deformations $\Delta = 2$, the parametric enhancement is logarithmic). The case $\Delta = 0$, which formally applies to a free scalar CFT with a $\phi^n$ deformation, is special and is treated separately – one finds that the bound (4) is replaced by $\tau_{\text{eq}} \gtrsim \frac{1}{T}\left(\frac{T}{\Lambda}\right)^{\frac{4}{n+2}}$. The slow thermalization in the high temperature limit is illustrated in Fig. 1.

Depending on microscopic details, some 2d QFTs may have the emergence of diffusive hydrodynamics before ultimately reaching KPZ universality – in these cases (4) still holds for the onset of diffusion (and in fact can be strengthened). This occurs for example in theories with many degrees of freedom $c_{\text{UV}} \gg 1$. Since KPZ universality arises from hydrodynamic fluctuations, which are suppressed at large $c_{\text{UV}}$, the bound (4) becomes weak in this limit;

however we show that causality still leads to a strong constraint on thermalization and the bulk viscosity, through the following bound

$$\frac{\zeta}{s} \lesssim (1-c_s)T\tau_{\text{eq}},\tag{6}$$

when $1-c_s \ll 1$.

In higher dimensions, QFTs thermalize slowly at high (low) temperatures if they become free in the UV (IR), a process which can be studied with Boltzmann kinetic theory. Instead, 2d QFTs *always* thermalize slowly in these regimes, regardless of the nature of the UV and IR CFTs; our results do not rely on a particle description.

This paper is organized as follows. In Sec. 2, we study the thermodynamics of 2d QFTs; we show that the dimensionless entropy density $s_o(T)$ defines a $C$-function, and use conformal perturbation theory to obtain the equation of state at high temperatures. We then turn to hydrodynamics in Sec. 3, explaining how KPZ universality emerges in 2d QFTs. Sec. 4 is devoted to the derivation of the bound (4). A number of extensions are discussed in Sec. 5, where we show that thermalization is also slow in the low temperature limit, and comment on higher dimensions, theories with additional global symmetries, and theories with a large number of degrees of freedom.

## 2 Thermodynamics

The equation of state of the QFT can be parametrized by the energy density $\varepsilon(T)$ or pressure $P(T)$, defined as the thermal expectation value of the stress tensor

$$\langle \mathcal{T}_{\mu\nu} \rangle_\beta = (\varepsilon + P)\delta_\mu^0 \delta_\nu^0 + P\eta_{\mu\nu},\tag{7}$$

where $\eta \equiv \text{diag}(-1,1)$ and $\beta = 1/T$ is the inverse temperature. We work in the thermodynamic limit, with volume $V \to \infty$. It is convenient to study the equation of state using the dimensionless entropy density, defined as

$$s_o(T) \equiv \frac{s}{T}, \qquad s = \frac{dP}{dT}.\tag{8}$$

One can show that in two dimensions, $s_o(T)$ is a $C$-function interpolating between the central charges of the UV and IR CFTs. First note that its high and low temperature limits are given by the Cardy formula

$$\lim_{T\to\infty} s_o(T) = \frac{\pi}{3}c_{\text{UV}}, \qquad \lim_{T\to 0} s_o(T) = \frac{\pi}{3}c_{\text{IR}}.\tag{9}$$

If the IR is gapped, $c_{\text{IR}} = 0$. Next, strict subluminality of sound propagation for nonintegrable flows implies that $s_o(T)$ is monotonic: the speed of sound is given by $c_s^2 = dP/d\varepsilon$, or, using $d\varepsilon = T ds$,

$$\frac{1}{c_s^2} = \frac{d\log s}{d\log T} = 1 + \frac{d\log s_o}{d\log T} > 1 \qquad \Longleftrightarrow \qquad s_o'(T) > 0.\tag{10}$$

This implies $c_{\text{UV}} > c_{\text{IR}}$, at least for nonintegrable RG flows. Note that this $C$-function is qualitatively different from the one introduced by Zamolodchikov [7], which involves vacuum two-point correlators of the stress tensor instead of thermal correlators. Subluminality of sound therefore offers an alternative proof of the 2d $C$-theorem in flows where hydrodynamics emerges.[1] A related thermodynamical $C$-function has been previously considered in the

---

[1]Even in integrable flows, turning on an irrelevant integrability breaking deformation should restore hydrodynamics, so that one expects $c_s^2 \le 1$ there as well. The $C$-function $s_o$ is illustrated in an integrable flow in Appendix A.2.

literature [8–12], which however is sensitive to the regulation scheme and requires fine-tuning the $T = 0$ vacuum energy density to zero. We comment further on the relation between these two $C$-functions in Appendix B. The advantage of the $C$-function proposed here, $s_o(T)$, is that it is insensitive to the vacuum energy density. In particular, one can measure it or calculate it at some specific temperature, without needing perfect knowledge of the vacuum energy or equivalently the pressure at $T = 0$ (should those be difficult to obtain).

The leading correction to the equation of state at high $T$ can be obtained from conformal perturbation theory in the UV CFT. Up to quadratic order in the deformation $\lambda$, the pressure is given by[2]

$$P = \frac{T}{V} \log Z$$
$$= \frac{\pi}{6} c_{UV} T^2 - \lambda \langle \mathcal{O} \rangle_\beta + \frac{\lambda^2}{2} \int_0^\beta d\tau \int dx \, \langle \mathcal{O}(\tau, x) \mathcal{O} \rangle_\beta + \cdots. \tag{11}$$

All correlation functions are evaluated in the UV CFT at finite temperature. In two dimensions, only global primaries in the identity multiplet can acquire thermal expectation values – since $\mathcal{O} \neq \mathbb{1}$ is a relevant scalar, the linear term vanishes. The quadratic term involves the thermal two-point function, which is fixed by conformal invariance[3]

$$\langle \mathcal{O}(x, \tau) \mathcal{O} \rangle_\beta = c_{UV} \left[ \frac{(\pi T)^2}{\sinh \frac{\pi}{\beta}(x + i\tau) \sinh \frac{\pi}{\beta}(x - i\tau)} \right]^\Delta. \tag{12}$$

Integrating over $x$ and $\tau$ gives [13, 15]:

$$\int_0^\beta d\tau \int dx \, \langle \mathcal{O}(x) \mathcal{O} \rangle_\beta = c_{UV} \pi (2\pi T)^{2\Delta - 2} \frac{\Gamma(1 - \Delta)\Gamma(\frac{\Delta}{2})^2}{\Gamma(\Delta)\Gamma(1 - \frac{\Delta}{2})^2}. \tag{13}$$

Note that for $1 < \Delta < 2$, a UV (OPE) divergence has been removed before performing the integral. Writing the pressure as

$$P = \frac{\pi}{6} c_{UV} T^2 \left[ 1 - \frac{\alpha_\Delta}{\Delta - 1} \left( \frac{\lambda}{T^{2-\Delta}} \right)^2 + \cdots \right], \tag{14}$$

with

$$\alpha_\Delta = 3(2\pi)^{2(\Delta-1)} \frac{\Gamma(2 - \Delta)\Gamma(\frac{\Delta}{2})^2}{\Gamma(\Delta)\Gamma(1 - \frac{\Delta}{2})^2} \geq 0, \qquad \text{(for } 0 < \Delta \leq 2), \tag{15}$$

one finds that the dimensionless entropy is

$$s_o = \frac{1}{T} \frac{dP}{dT} = \frac{\pi}{3} c_{UV} \left[ 1 - \alpha_\Delta \left( \frac{\lambda}{T^{2-\Delta}} \right)^2 + \cdots \right]. \tag{16}$$

Since only temperature independent UV divergences were removed in (13), these do not contribute to the entropy so that $s_o$ is UV insensitive as anticipated above (and in contrast to

---

[2]This expansion is free of IR divergences as long as $\Delta > 0$. There are power-law UV divergences when $\Delta > 1$, which can be removed with temperature independent counterterms. Finally, there are physical (temperature-dependent) logarithmic UV divergences at order $\lambda^n$ in conformal perturbation theory when $\Delta = 2 - \frac{2}{n}$, for $n = 1, 2, 3, \ldots$ [13, 14]. Examples with $n = 1$ and 2 are discussed in Appendix A – for $n > 2$ these terms with logarithmic enhancements do not control the leading correction to the equation of state.

[3]We have normalized the deforming operator as $\lim_{x \to 0} \mathcal{O}(x) \mathcal{O} \sim c_{UV}/x^{2\Delta}$ rather than $1/x^{2\Delta}$. This is unimportant when $c_{UV} \sim 1$, but when $c_{UV} \gg 1$ it guarantees that the relative correction to the equation of state is order 1 at temperatures $T \sim \Lambda \equiv \lambda^{1/(2-\Delta)}$.

a previously proposed thermodynamic $C$-function, see Appendix B). The correction to $s_o$ is negative as expected from Eq. (10) – degrees of freedom are lost as $T$ is decreased. Sound therefore propagates subluminally

$$c_s^2 = 1 - 2(2-\Delta)\alpha_\Delta \left( \frac{\lambda}{T^{2-\Delta}} \right)^2 + \cdots < 1\,, \tag{17}$$

and the speed of sound $c_s$ approaches the conformal value $c_s = 1$ from below as the temperature is increased. These expressions diverge for $\Delta = 0$ – although no unitary CFT has a nontrivial operator of vanishing dimension, the free scalar with a relevant deformation $\phi^n$ formally realizes this possibility. This case is treated in Appendix A. For marginally relevant deformations $\Delta = 2$ this correction vanishes. The first correction to the equation of state arises at order $\lambda^3$, and a logarithmically enhanced contribution arises at order $\lambda^4$ [13] – resumming the leading large logarithms[4] and taking the high temperature limit leads to a correction

$$\delta c_s \sim \frac{\delta s_o}{s_o} \sim -\frac{1}{(\log T/\Lambda)^3}\,. \tag{18}$$

## 3  Hydrodynamics

The real time dynamics of interacting systems at finite temperature is described by hydrodynamics at sufficiently late times $t \gg \tau_{\rm eq}$ (see [16] for an introduction to relativistic hydrodynamics). The equilibration time $\tau_{\rm eq}$ can be interpreted as the UV cutoff of the effective hydrodynamics description. In this regime, the only surviving excitations are those associated with conserved quantities. For QFTs with no internal continuous global symmetry, these are the stress tensor components $\mathcal{T}^{00}$ and $\mathcal{T}^{0x}$, whose lifetimes are protected by the conservation laws

$$\partial_\mu \mathcal{T}^{\mu\nu} = 0\,. \tag{19}$$

We can parametrize these long-lived collective excitations (i.e., sound modes) either directly in terms of the local energy density $\varepsilon(x)$ and momentum density $\pi(x) \equiv \mathcal{T}^{0x}$, or alternatively, in terms of the local temperature $T(x)$ and velocity $u^\mu(x)$, satisfying $u_\mu u^\mu = -1$. The latter choice is often useful for making the Lorentz invariance of the underlying QFT more manifest.

At late times in the thermal state, correlation functions of any neutral operator of the QFT can be expanded in terms of these hydrodynamic degrees of freedom in a derivative expansion – these operator matching equations are called *constitutive relations*. For the stress tensor, the most general constitutive relation up to first order in derivatives is

$$\mathcal{T}^{\mu\nu} = (\varepsilon + P)u^\mu u^\nu + P\eta^{\mu\nu} - \zeta \partial_\lambda u^\lambda (\eta^{\mu\nu} + u^\mu u^\nu) + O(\partial^2)\,, \tag{20}$$

where $\zeta$ is the bulk viscosity.[5] Note that there is no shear viscosity in two dimensions. For notational simplicity, we have suppressed the dependence on position, but it is important to note that the densities on the right-hand side of (20) are all *local* (i.e., $\varepsilon(x)$ and $P(x)$).

For $d > 2$, all interactions for the hydrodynamic modes are technically irrelevant and the late-time dynamics in generic QFTs is governed by diffusion, such that the collective

---

[4]More explicitly, the $\beta$ function for the coupling $\lambda$ is $\beta_\lambda = (2-\Delta)\lambda - C\lambda^2$, with $C = C_{\mathcal{O}\mathcal{O}\mathcal{O}}$ being an OPE coefficient, so the running coupling $\lambda(T) \sim \frac{1}{\lambda_0^{-1} - C \log(T/T_0)}$ at $\Delta = 2$. Then $\delta P \sim \langle \mathcal{T}^\mu_\mu \rangle \sim \beta_\lambda \langle \mathcal{O} \rangle \sim \beta_\lambda \lambda \langle \mathcal{O}\mathcal{O} \rangle$, and $\beta_\lambda \lambda \sim \frac{1}{C^2 \log^3 T}$ after using the running coupling at large $T$.

[5]A term $\mathcal{T}^{\mu\nu} \supset a_1 u^{(\mu}\partial^{\nu)}T$ was removed with a field redefinition $u^\mu \to u^\mu + \delta u^\mu$, and similarly for $\mathcal{T}^{\mu\nu} \supset u^\mu u^\nu (a_2 \partial \cdot u + a_3 u \cdot \partial T)$ using $T \to T + \delta T$. Finally, the leading equations of motion $u_\mu \partial_\nu \mathcal{T}^{\mu\nu} = 0$ were used to absorb a term $\mathcal{T}^{\mu\nu} \supset a_4(\eta^{\mu\nu} + u^\mu u^\nu)u \cdot \partial T$ in the bulk viscosity, up to higher derivative terms.

excitations largely propagate freely, but dissipate due to the viscosities. All late-time thermal correlators are therefore determined by the two-point functions of the densities $\varepsilon(x)$, $\pi(x)$ (or equivalently $T(x)$, $u^\mu(x)$). However, as we will show below, in two dimensions there are relevant interactions, which significantly alter the late-time dynamics. Before discussing the effects of interactions in 2d, though, we briefly review the usual linearized analysis applicable in higher dimensions.

The typical approach is to expand the densities around their thermal expectation values,

$$\varepsilon(x) = \varepsilon + \delta\varepsilon(x), \qquad \pi(x) = 0 + \delta\pi(x), \tag{21}$$

or equivalently,

$$T(x) = T + \delta T(x), \qquad u^\mu(x) = \frac{1}{\sqrt{1 - \delta v^2(x)}} \begin{pmatrix} 1 \\ \delta v(x) \end{pmatrix}, \tag{22}$$

with the relation between these two parametrizations given by

$$\delta\varepsilon(x) \simeq \frac{d\varepsilon}{dT} \delta T(x), \qquad \delta\pi(x) \simeq (\varepsilon + P) \delta v(x). \tag{23}$$

Thermal correlation functions can be obtained by inserting (20) into (19), expanding around the thermal background to linear order in the fluctuations $\delta\varepsilon$, $\delta\pi$ (or $\delta T$, $\delta v$), and solving for the densities in terms of the sources [17], see [16] for details. This procedure leads to the following retarded Green's function for momentum density:

$$G^R_{\mathcal{T}_{0x}\mathcal{T}_{0x}}(\omega, k) \simeq sT \frac{c_s^2 k^2 - iD\omega k^2}{c_s^2 k^2 - \omega^2 - iD\omega k^2}, \tag{24}$$

with diffusion constant[6] given by

$$D \equiv \frac{\zeta}{sT}. \tag{25}$$

The Fourier transform of the Wightman function can be obtained from (24) using the general relation

$$\langle AA \rangle(\omega, k) = \frac{2}{1 - e^{-\beta\omega}} \operatorname{Im} G^R_{AA}(\omega, k) \simeq \frac{2}{\beta\omega} \operatorname{Im} G^R_{AA}(\omega, k), \tag{26}$$

where in the last step we used the fact that $\beta\omega \ll 1$ in the hydrodynamic regime. For the energy density, this gives

$$\begin{aligned}
\langle \mathcal{T}_{0x}\mathcal{T}_{0x} \rangle(\omega, k) &\simeq 2T^2 s \frac{D\omega^2 k^2}{(\omega^2 - c_s^2 k^2)^2 + (D\omega k^2)^2} \\
&\simeq \frac{sT^2}{Dk^2} \frac{\omega}{c_s k} \left[ g_{\text{diff}}\left( \frac{\omega - c_s k}{\frac{1}{2}Dk^2} \right) - g_{\text{diff}}\left( \frac{\omega + c_s k}{\frac{1}{2}Dk^2} \right) \right].
\end{aligned} \tag{27}$$

In the second line we have separated the contributions from the two poles and expressed the result in terms of a 'diffusion scaling function' $g_{\text{diff}}(x) \equiv \frac{2}{1+x^2}$. In two dimensions, all other components of the stress tensor Green's function can be obtained from (27) through the Ward identity [18] $p^\mu \langle \mathcal{T}_{\mu\nu}\mathcal{T}_{\lambda\rho}\rangle(p) = $ contact terms.

However, Eq. (27) does not correctly describe stress tensor thermal correlators in generic two-dimensional QFTs. We have incorrectly assumed in Eq. (21) that hydrodynamic fluctuations could be linearized around equilibrium. It is well known, but perhaps not entirely appreciated

---

[6]While $D$ is more accurately a sound attenuation (or damping) rate, in a slight abuse of language we will refer to it as a diffusion constant.

in the high-energy literature, that sound modes in two dimensions lead to large hydrodynamic fluctuations. In other words, there are relevant interactions, which lead to a breakdown of diffusive spreading of ballistic modes and trigger a flow to a new 'dissipative fixed point' in the Burgers-Kardar-Parisi-Zhang (KPZ) universality class [19–22]. Let us review how this arises in the present context. Expanding the hydrodynamic equations (19) and (20) in perturbations gives the equations of motion

$$0 = \partial_t \delta\varepsilon + \nabla\delta\pi + \frac{1}{\varepsilon + P}\partial_t(\delta\pi)^2 + \cdots, \tag{28a}$$

$$0 = \partial_t \delta\pi + P'(\varepsilon)\nabla\delta\varepsilon - D\nabla^2\delta\pi + \frac{1}{2}P''(\varepsilon)\nabla(\delta\varepsilon)^2 + \frac{1}{\varepsilon + P}\nabla(\delta\pi)^2 + \cdots, \tag{28b}$$

where $\nabla \equiv \partial_x$. The ellipses $\cdots$ contain terms that are higher order in derivatives or fluctuations $O(\partial\delta^3, \partial^2\delta^2)$. To leading order, we can factorize these coupled differential equations by introducing the right- and left-moving modes

$$\pi_\pm \equiv \delta\pi \pm c_s\delta\varepsilon, \tag{29}$$

with $c_s \equiv \sqrt{P'(\varepsilon)}$, obtaining

$$0 = \partial_t\pi_+ + c_s\nabla\pi_+ - \frac{1}{2}D\nabla^2\pi_+ + \frac{1}{2}\kappa\nabla\pi_+^2 + \cdots, \tag{30a}$$

$$0 = \partial_t\pi_- - c_s\nabla\pi_- - \frac{1}{2}D\nabla^2\pi_- + \frac{1}{2}\kappa\nabla\pi_-^2 + \cdots, \tag{30b}$$

with coefficient for the nonlinear term

$$\kappa = \frac{1}{2}\left(\frac{1}{2}\frac{P''(\varepsilon)}{P'(\varepsilon)} + \frac{1 - P'(\varepsilon)}{\varepsilon + P}\right). \tag{31}$$

Here we have ignored interactions between the two modes as these are irrelevant [22][7]. The linear terms for these modes reproduce the two poles $\omega \simeq \pm c_s k - \frac{i}{2}Dk^2$ in Eq. (27).

The fact that interactions are relevant in 2d follows from a simple scaling argument. Let us focus on the right-moving $\pi_+$ mode for concreteness and work in the coordinates $x' = x - c_s t$, $t' = t$ where the equation of motion simplifies to

$$0 = \partial_{t'}\pi_+ - \frac{1}{2}D\nabla^2\pi_+ + \frac{1}{2}\kappa\nabla\pi_+^2 + \cdots. \tag{32}$$

In these coordinates, the linearized density correlator from (27) is diffusive,

$$\langle\pi_+\pi_+\rangle(t, x' = 0) \sim \frac{sT^2}{|Dt|^{1/2}}, \tag{33}$$

showing that densities scale as $\pi_+ \sim \omega'^{1/4} \sim k^{1/2}$. The interaction term $\nabla\pi_+^2 \sim k^2$ is therefore *more relevant* than the diffusive term $\nabla^2\pi_+ \sim k^{5/2}$. At late times, hydrodynamic modes in 2d are therefore strongly-interacting, unlike in higher dimensions.

The hydrodynamics that arises from Eq. (32) has been thoroughly studied (see [22] for a review). Rather than being diffusive, it is described by the KPZ universality class: the correlation function in Eq. (27) is replaced by[8]

$$\langle\mathcal{T}_{0x}\mathcal{T}_{0x}\rangle(\omega, k) \simeq \frac{sT^2}{2\mathcal{D}k^{3/2}}\frac{\omega}{c_s k}\left[g_{\text{KPZ}}\left(\frac{\omega - c_s k}{\mathcal{D}|k|^{3/2}}\right) - g_{\text{KPZ}}\left(\frac{\omega + c_s k}{\mathcal{D}|k|^{3/2}}\right)\right]. \tag{34}$$

---

[7]Indeed, in coordinates where the $\pi_+$ correlator scales as (33), $\pi_-$ decays exponentially $\langle\pi_-\pi_-\rangle(t, x' = 0) \sim \frac{e^{-t/D}}{\sqrt{Dt}}$. This factorization, arising due to the peculiarities of 1+1d kinematics, is reminiscent of the holomorphic factorization of CFTs.

[8]This expression holds near the singularities $\omega \simeq \pm c_s k$, where the correlator is large. Away from the singularities, the decoupling between $\pi_+$ and $\pi_-$ is no longer valid (c.f. footnote 7); the solution to the resulting coupled KPZ equation is not known in general [22]. The correlator (34) may therefore be multiplied by a function of $\frac{\omega}{c_s k}$ equal to unity when $\omega = \pm c_s k$. Its $k \to 0$ limit must however take the form in (34) because $\mathcal{T}_{0x}$ is a conserved quantity. We will only use these two properties of this function in what follows.

Here $g_{\text{KPZ}}$ is the KPZ scaling function and the parameter $\mathcal{D}$ is fixed in terms of the equation of state through the interaction $\kappa$ in (31)

$$\mathcal{D} = \sqrt{T \chi_{\pi_+ \pi_+}} |\kappa| = \sqrt{2sT^2} |\kappa| \,, \tag{35}$$

where the susceptibility associated with $\pi_+$ was evaluated using (29) and (23):

$$\chi_{\pi_+ \pi_+} = \chi_{\pi\pi} + c_s^2 \chi_{\varepsilon\varepsilon} \equiv \frac{d\pi}{d\nu} + c_s^2 T \frac{d\varepsilon}{dT} = 2sT \,. \tag{36}$$

The KPZ scaling function is not known analytically but known numerically to high precision [23]; it has the following properties

$$\int \frac{dx}{2\pi} g_{\text{KPZ}}(x) = 1 \,, \qquad \lim_{x \to \infty} g_{\text{KPZ}}(x) = \frac{2a}{x^{7/3}} \,, \qquad \lim_{x \to 0} g_{\text{KPZ}}(x) = b \,, \tag{37}$$

with $a \approx 0.417816$ and $b \approx 3.43730$. We comment on a few similarities and differences between KPZ and diffusive correlators. Both the KPZ hydrodynamic correlator (34) and the diffusive one (27) saturate the sum rules at small wavevector[9] $k \ll T$

$$\int \frac{d\omega}{\pi} \frac{1}{\omega} \operatorname{Im} G^R_{\mathcal{T}_{0x} \mathcal{T}_{0x}}(\omega, k) = \operatorname{Re} G^R_{\mathcal{T}_{0x} \mathcal{T}_{0x}}(\omega = 0, k) = sT \,, \tag{38a}$$

$$\int \frac{d\omega}{\pi} \frac{1}{\omega} \operatorname{Im} G^R_{\mathcal{T}_{00} \mathcal{T}_{00}}(\omega, k) = \operatorname{Re} G^R_{\mathcal{T}_{00} \mathcal{T}_{00}}(\omega = 0, k) = T \frac{d\varepsilon}{dT} \,, \tag{38b}$$

as can be checked by using (26) and $\int \frac{dx}{2\pi} g_{\text{KPZ}}(x) = \int \frac{dx}{2\pi} g_{\text{diff}}(x) = 1$. However, in the KPZ regime, the bulk viscosity is singular at low frequencies

$$\zeta(\omega) \equiv \lim_{k \to 0} \frac{1}{\omega} \operatorname{Im} G^R_{\mathcal{T}_{xx} \mathcal{T}_{xx}}(\omega, k) = sT \frac{\frac{7}{3} a \mathcal{D}^{4/3}}{\omega^{1/3}} \,, \tag{39}$$

with $a$ given below Eq. (37). Similar behavior has been found for heat or charge conductivities in 1+1d non-relativistic systems [21,24,25]; in the present relativistic context the heat conductivity vanishes because the energy current is a conserved density.

It is a striking feature of dissipation in the KPZ universality class that real time correlation functions (34) are entirely fixed in terms of thermodynamic quantities, in contrast to (27) which involves an unknown diffusion constant. In relativistic two dimensional QFTs at high temperatures $T \gg \lambda^{1/(2-\Delta)} \equiv \Lambda$, the thermodynamic results from Sec. 2 can be used in (31) and (35) to obtain the KPZ transport parameter

$$\mathcal{D}\sqrt{T} \simeq \sqrt{\frac{6}{\pi c_{\text{UV}}} (2 - \Delta)(3 - \Delta) \alpha_\Delta \left( \frac{\Lambda}{T} \right)^{2(2-\Delta)}} + \cdots \,. \tag{40}$$

Breaking of conformal invariance by the deformation (1) therefore has two consequences: First, it opens a window for hydrodynamics by allowing for $c_s < 1$, and second, it produces nonlinearities in the hydrodynamic equations of motion (30), ultimately leading to large hydrodynamic fluctuations and dissipation in the KPZ universality class with transport parameter given by (40).

In certain situations (notably when $c_{\text{UV}} \gg 1$), $\mathcal{D}$ is small so that there may be an intermediate window with diffusive hydrodynamics, before a cross-over time when the system ultimately enters KPZ universality. This time scale can be found by balancing the last two terms in (32),

$$\tau_{\text{cross-over}} \sim \frac{D^3}{\mathcal{D}^4} \sim \frac{1}{T} c_{\text{UV}}^2 (DT)^3 \left( \frac{T}{\Lambda} \right)^{8(2-\Delta)} \,, \tag{41}$$

where in the last step we used the high temperature expansion $T \gg \Lambda$ for $\mathcal{D}$ in (40).

---

[9]One can estimate the contribution to the sum rule from large frequencies using the CFT correlator $\frac{1}{T} \int_T^\infty d\omega \langle \mathcal{T}_{0x} \mathcal{T}_{0x} \rangle \sim c_{\text{UV}} k^2 \ll s_o T^2$ which is indeed suppressed compared to (38) when $k \ll T$.

# 4   Derivation of the Bound

Let us start by reviewing (a suitable generalization of) the thermalization bound of Ref. [2]. A retarded Green's function $G^R(t,x)$ involving a hydrodynamic mode with dispersion relation $\omega \simeq c_s k - i\mathcal{D}k^z$ is exponentially suppressed except for $||x| - c_s t| \lesssim (\mathcal{D}t)^{1/z}$. At early times and for $z > 1$, these points are outside of the lightcone $|x| = t$, where the commutator in $G^R$ vanishes by causality. These two statements can be reconciled only if hydrodynamics emerges at later times, leading to the bound in Eq. (3) on $\tau_{\text{eq}}$. For the hydrodynamic correlators of 2d QFTs in Eq. (34), KPZ dissipation with $z = \frac{3}{2}$ leads to

$$\tau_{\text{eq}} \gtrsim \tau_{\text{KPZ bound}} \equiv \frac{\mathcal{D}^2}{(1-c_s)^3} = \frac{1}{T}\frac{1}{s_o}\left[\frac{1}{2}\frac{1}{(1-c_s)^3}\left(\frac{d\log c_s}{d\log s} + 1 - c_s^2\right)^2\right], \qquad (42)$$

where we expressed $\mathcal{D}$ in terms of the equation of state using (35) and (31). This is a bound on the equilibration time of 2d QFTs involving only equilibrium thermodynamic quantities. QFTs that do not have a large number of degrees of freedom have $s_o \sim 1$ and therefore satisfy the 'Planckian' bound $\tau_{\text{eq}} \gtrsim \frac{1}{T}$ [6], as long as the equation of state leads to a $\gtrsim O(1)$ quantity in square brackets in (42). At high temperatures $T \gg \Lambda$, one can in fact show that this quantity is always large, so that 2d QFTs (with $s_o \sim 1$) always thermalize much more slowly than the Planckian time $\frac{1}{T}$ in this regime. The same is true at low temperatures, see Sec. 5.2. In the remainder of this section, we focus on the high temperature limit $T \gg \Lambda$. Using the expressions for $c_s$ (17) and $\mathcal{D}$ (40) obtained from conformal perturbation theory and dropping numerical coefficients, the bound for $T \gg \Lambda$ becomes

$$\tau_{\text{eq}} \gtrsim \tau_{\text{KPZ bound}} \sim \frac{1}{T}\frac{1}{(2-\Delta)\alpha_\Delta c_{\text{UV}}}\left(\frac{T}{\Lambda}\right)^{2(2-\Delta)}, \qquad (0 < \Delta < 2). \qquad (43)$$

This is the result quoted in Eq. (4). For marginally relevant deformations ($\Delta = 2$), the logarithmic corrections to the equation found in (18) give

$$\tau_{\text{eq}} \gtrsim \tau_{\text{KPZ bound}} \sim \frac{1}{T}\frac{1}{c_{\text{UV}}}\left(\log\frac{T}{\Lambda}\right)^3, \qquad (\Delta = 2). \qquad (44)$$

The case $\Delta = 0$, which applies to a massless scalar with $\phi^n$ deformation, is discussed in Appendix A.3 – a result similar to (43) holds.

In certain situations, diffusive hydrodynamics will emerge first, before ultimately settling to KPZ hydrodynamics after the cross-over time (41). In this situation it is more pertinent to define thermalization as the emergence of diffusive hydrodynamics. The bound on the emergence of diffusion can be found by setting $z = 2$ in (3), leading to

$$\tau_{\text{eq}} \gtrsim \tau_{\text{diff. bound}} \equiv \frac{D}{(1-c_s)^2} = \frac{D}{D_{\text{cr}}}\tau_{\text{KPZ bound}}, \qquad (45)$$

where in the last step we expressed the result in terms of a critical diffusive constant which in the high temperature limit reads

$$TD_{\text{cr}} \equiv \frac{T\mathcal{D}^2}{1-c_s} \sim \left(\frac{\Lambda}{T}\right)^{2(2-\Delta)}\frac{(2-\Delta)\alpha_\Delta}{c_{\text{UV}}}. \qquad (46)$$

In this notation, the cross-over time (41) is

$$\tau_{\text{cross-over}} \sim \left(\frac{D}{D_{\text{cr}}}\right)^3 \tau_{\text{KPZ bound}}. \qquad (47)$$

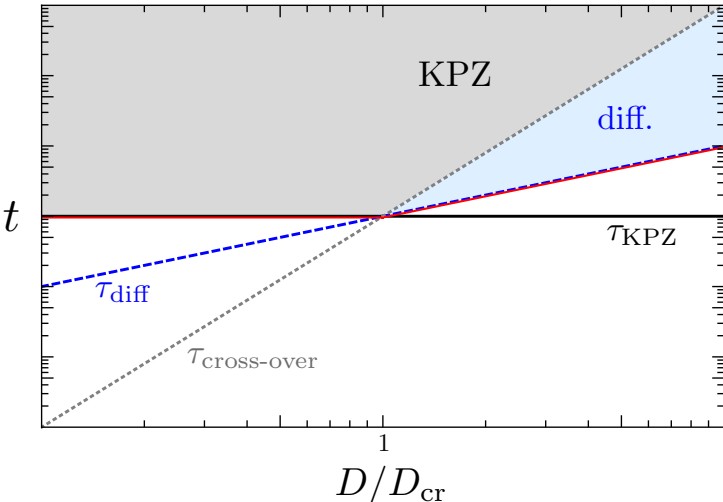

Figure 2: Hydrodynamic timescales as a function of the diffusion constant $D$, for $T \gg \Lambda$. For $D \lesssim D_{\mathrm{cr}}$, hydrodynamics is governed by KPZ (gray region), with the equilibration time bounded by $\tau_{\mathrm{KPZ\ bound}}$ (black line). For $D \gtrsim D_{\mathrm{cr}}$, the equilibration time satisfies a stronger diffusive bound $\tau_{\mathrm{diff.\ bound}}$ (blue dashed line); hydrodynamics is initially governed by diffusion (blue region) before transitioning to KPZ at $\tau_{\mathrm{cross\text{-}over}}$ (gray dotted line). The equilibration time therefore satisfies the bound $\tau_{\mathrm{eq}} \gtrsim \max(\tau_{\mathrm{KPZ\ bound}}, \tau_{\mathrm{diff.\ bound}})$ (red line).

The fate of thermalization then depends on how the diffusion constant $D$ compares to the value (46), as shown in Fig. 2. If $D \lesssim D_{\mathrm{cr}}$, then the cross-over to KPZ physics (47) happens before diffusion can kick in; the only hydrodynamic regime is KPZ and the bound (43) is unchanged. If instead $D \gtrsim D_{\mathrm{cr}}$, diffusion can emerge before KPZ physics – the appropriate bound is then (45), which is stronger than the KPZ bound (43) by a factor of $D/D_{\mathrm{cr}}$

$$\tau_{\mathrm{eq}} \gtrsim \tau_{\mathrm{diff.\ bound}} \sim \frac{D}{D_{\mathrm{cr}}} \tau_{\mathrm{KPZ\ bound}}. \tag{48}$$

In this situation, the cross-over time (47) determines when the system ultimately settles to KPZ dissipation.

## 4.1 Identification of the long-lived modes

The slow thermalization of 2d QFTs suggests the existence of long-lived non-hydrodynamic modes.[10] At high temperatures, the physics is governed by a CFT; we start by considering the case where that CFT is free, and then discuss the general case. If the UV CFT is free (i.e. the QFT is asymptotically free) these long-lived modes are particles, and – ignoring IR issues which we comment on further at the end of this section – thermalization can be studied perturbatively using Boltzmann kinetic theory as in higher dimensions (see, e.g., [26]). In this case the equilibration time can be estimated from a thermally averaged cross-section $\langle \sigma \rangle \sim \lambda^2 / T^{2(2-\Delta)}$ as

$$\tau_{\mathrm{eq}} \sim \frac{1}{sv\langle\sigma\rangle} \sim \frac{1}{T}\frac{1}{c_{\mathrm{UV}}}\frac{T^{2(2-\Delta)}}{\lambda^2}, \tag{49}$$

showing that the bound (43) is parametrically saturated[11]. However, the bound (43) holds for any UV CFTs, not only for those with a quasiparticle description. In the following, we give an

---

[10]We thank Sean Hartnoll and Tom Hartman for discussions that led to the argument presented in this section.

[11]KPZ dissipation can moreover be derived in certain weakly coupled 2d QFTs [27, 28].

argument for the saturation of the bound that does not rely on kinetic theory.

When thermalization is impeded by a single slow operator, the equilibration time can be obtained in a systematic expansion using the *memory function* formalism to compute the small relaxation rate of the slow operator [29, 30] (see [31] for a recent review). In the present situation, we expect an infinite tower of operators to be approximately conserved at high temperatures due to the Virasoro symmetry (or KdV charges) in the UV. However one can nevertheless estimate the equilibration time by choosing one of them (we will comment below on situations where this estimate is too naive). The first KdV charge whose conservation is broken by the relevant deformation $\mathcal{O}$ is [32]

$$Q_3 \equiv \int dx : \mathcal{T}^2 :, \qquad \partial_t Q_3 \sim \lambda \int dx \, \widetilde{\mathcal{O}}, \tag{50}$$

where $\widetilde{\mathcal{O}}$ is a level-3 Virasoro descendent of $\mathcal{O}$ (only the global primary $\widetilde{\mathcal{O}} \supset L_{-3}\mathcal{O}$ will contribute below). The memory function formalism roughly amounts to modeling the correlator of the long lived operator as $G_{Q_3 Q_3}^R(\omega) \simeq \chi_{Q_3 Q_3} \frac{\Gamma}{-i\omega + \Gamma}$ and working perturbatively in its rate relaxation $\Gamma$ (or the deformation $\lambda$) to obtain the following Kubo formula [31]

$$\Gamma \simeq \frac{1}{\chi_{Q_3 Q_3}} \lim_{\omega \to 0} \frac{1}{2T} \langle \dot{Q}_3 \dot{Q}_3 \rangle_\beta(\omega) \sim \lambda^2 \frac{1}{\chi_{Q_3 Q_3}} \lim_{\omega \to 0} \frac{1}{T} \langle \widetilde{\mathcal{O}} \widetilde{\mathcal{O}} \rangle_\beta(\omega). \tag{51}$$

Here the $\lambda \to 0$ limit is taken before $\omega \to 0$, so that the Wightman functions (related to the retarded Green's function by (26)) are evaluated in the unperturbed CFT. Since $\chi_{Q_3 Q_3} \sim c_{\mathrm{UV}}^2 T^6$ and $\langle \widetilde{\mathcal{O}} \widetilde{\mathcal{O}} \rangle \sim c_{\mathrm{UV}}^2 T^{2\Delta + 4}$ one finds

$$\tau_{\mathrm{eq}} \sim \frac{1}{\Gamma} \sim \frac{1}{T} \left( \frac{T}{\Lambda} \right)^{2(2-\Delta)}. \tag{52}$$

This suggests that, at least for theories with $c_{\mathrm{UV}} \sim 1$, the bound (43) is parametrically saturated. We expect this argument to hold in generic 2d QFTs, with no small parameter.

There are three situations where the analysis is slightly more subtle: (i) In integrable or nearly integrable flows (such as the one considered in Appendix A.2), there exist operators that are much longer lived than the spin-4 current in (50), leading to a further parametric enhancement in the thermalization time, so that the bound (43) is not tight. (ii) In theories with a large number of degrees of freedom $c_{\mathrm{UV}} \gg 1$, although the operator (50) is very long-lived it decouples from the 'single-trace' sector[12] and therefore does not preclude thermalization. We in fact expect certain $c_{\mathrm{UV}} \gg 1$ theories to thermalize fast, $\tau_{\mathrm{eq}} \sim \frac{1}{T}$, see Sec. 5.4. (iii) Finally, IR divergences at finite temperature can make the perturbative expansion more subtle. While some of these divergences can be resolved within perturbation theory by resumming hard thermal loops [34, 35], others are due to truly non-perturbative dynamics, such as the 'Linde problem' at the scale $\sim g^2 T$ in QCD [36]. This breakdown of perturbation theory also affects real time quantities [37–41]. In Appendix A.3, we study a 2d QFT – the free scalar with a $\phi^n$ deformation – where such nonperturbative dynamics already arises in the first correction to the pressure, and in the leading contribution to transport quantities. For example, for a deformation $\lambda \phi^4$ at high temperatures $T \gg \sqrt{\lambda}$ the pressure is $P - \frac{\pi}{6}T^2 \sim -\lambda^{1/3} T^{4/3}$ and our thermalization bound becomes $T\tau_{\mathrm{eq}} \gtrsim T^{2/3}/\lambda^{1/3}$.

It would be interesting to study the dynamics of these slow excitations more systematically, and we encourage the community to do so, perhaps in the context of 'generalized hydrodynamics' with small relaxation rates (see, e.g., Refs. [42–44]) applied to the KdV charges.

---

[12]More precisely, in the large $c_{\mathrm{UV}}$ limit, $Q_3$ and all the other KdV charges $Q_{2k-1}$ become redundant with the conserved energy from the stress tensor $\mathcal{T}$, up to $1/c_{\mathrm{UV}}$ corrections; see e.g. [33].

## 5 Extensions

### 5.1 Higher dimensions

Although Eq. (3) holds in any dimension $d$, in $d > 2$ it is not as constraining since the speed of sound does not approach the lightcone at high temperatures $\lim_{T\to\infty} c_s^2 = \frac{1}{d-1}$, and our bound (4) does not hold – we therefore have no reason to doubt the expectation that generic strongly interacting QFTs in $d > 2$ have 'Planckian' thermalization $\tau_{\text{eq}} \sim 1/T$ [6].

However one can still study the leading corrections to the high $T$ equation of state using conformal perturbation theory as in Sec. 2, although this expansion is only controlled if the UV CFT has a finite thermal mass (Appendix A.3 shows a simple example where the expansion is not controlled, see also [45] for a related discussion). If the relevant deformation breaks a symmetry, $\mathcal{O}$ does not have a thermal expectation value in the UV so that like in $d = 2$ the linear $\lambda$ term in Eq. (11) vanishes. The leading correction to the entropy density is therefore negative $\propto -\lambda^2$ as in Eq. (16), and the conformal value of the speed of sound is approached from below as the temperature is increased. This was found in 4d holographic models in [46,47]; we have shown more generally that this result is a consequence of conformal perturbation when the deformation does not have a thermal expectation value $\langle\mathcal{O}\rangle_\beta = 0$. Instead, when the relevant deformation breaks no symmetry of the UV CFT, it should acquire a thermal expectation value in the UV – in this case the leading correction to the pressure is instead

$$\delta P \simeq -\lambda\langle\mathcal{O}\rangle_\beta = -\lambda b_{\mathcal{O}} T^\Delta, \tag{53}$$

where generically one expects $b_{\mathcal{O}} = O(1)$ [48,49]. The correction to the speed of sound at high temperatures is then

$$\frac{1}{c_s^2} \simeq (d-1) + \Delta(d-\Delta)\frac{b_{\mathcal{O}}}{b_T}\frac{\lambda}{T^{d-\Delta}} + O\Big(\frac{\lambda^2}{T^{2(d-\Delta)}}\Big), \tag{54}$$

where we wrote $b_T \equiv \lim_{T\to\infty} s_o(T)$. This correction can have either sign, depending on the sign of $\lambda$, implying that the speed of sound can approach the conformal value either from below or above in $d > 2$ (for this same reason, $s_o(T)$ cannot be a $C$-function in $d > 2$ [10,12]). As an explicit example, one can consider the 3d $O(2)$ CFT with a mass deformation of either sign $\pm m^2\phi^2$.

### 5.2 Irrelevant deformations and low temperature limit

In this work we have focused on the high temperature limit of QFTs obtained from UV CFTs with a relevant deformation. Similar results apply to the low temperature limit of two-dimensional effective field theories (EFT), or CFTs deformed by irrelevant operators. One interesting difference arises because the scalar deformation $\mathcal{O}$ can now be a Virasoro descendant of the identity. In this case $\mathcal{O}$ acquires a thermal expectation value in the CFT, and the leading correction to the equation of state is linear in the deformation (as happens for $d > 2$, see Eq. (53)). The lightest such scalar is $\mathcal{O} = \mathcal{T}\bar{\mathcal{T}}$. In the context of effective field theory, one expects all irrelevant operators to be present as deformations of the CFT, so that the theory is described by

$$S_{\text{QFT}} = S_{\text{CFT}} + \sum_i \lambda_i \int d^2x\, \mathcal{O}_i - \lambda_{\mathcal{T}\bar{\mathcal{T}}} \int d^2x\, \mathcal{T}\bar{\mathcal{T}}, \tag{55}$$

where the $\mathcal{O}_i$ have dimension $2 \le \Delta_i$ and we have separated the $\mathcal{T}\bar{\mathcal{T}}$ term; although it is not the lightest irrelevant deformation, its contribution is enhanced because it is only linear in

coupling, and the correction to the speed of sound at low temperatures takes the form

$$\frac{1}{c_s^2} \simeq 1 - \sum_i \alpha_{\Delta_i} \lambda_i^2 T^{2(\Delta_i-2)} + \lambda_{\mathcal{T}\bar{\mathcal{T}}} c_{\text{IR}} T^2 + \cdots , \tag{56}$$

(we have absorbed positive numbers in the couplings $\lambda$). If there are irrelevant deformations with dimension $2 \leq \Delta_i < 3$, these will control the leading correction to the equation of state, and will give positive contributions to the $C$-function $s_o(T)$ and $\frac{1}{c_s^2}$, as expected by causality and mirroring the high temperature limit studied in Sec. 2 (notice that now $\alpha_\Delta < 0$ for $2 < \Delta < 3$, c.f. Eq. (15)).

However, if there is no irrelevant scalar of dimension $2 \leq \Delta_i < 3$ (or if their coefficients $\lambda_i$ are fine-tuned to zero), the leading correction to the equation of state seems to be non-sign-definite. Causality then requires the coefficient $\lambda_{\mathcal{T}\bar{\mathcal{T}}}$ to be positive in any QFT with a Lorentz invariant UV completion. When $S_{\text{CFT}}$ is a free scalar, this is a well known result [50]. In the context of the $\mathcal{T}\bar{\mathcal{T}}$ deformation, the relation between superluminal sound and the 'wrong sign' of $\lambda_{\mathcal{T}\bar{\mathcal{T}}}$ is also well known [51–53]. Our argument however does not require integrability: $\lambda_{\mathcal{T}\bar{\mathcal{T}}}$ must be positive even in the presence of an infinite tower of irrelevant terms in (55) with dimension $\Delta_i > 3$. In fact, if the leading irrelevant scalar has dimension $3 < \Delta_i < 4$, its contribution to Eq. (56) is instead *negative*, because now $\alpha_{\Delta_i} > 0$. In this case the $\mathcal{T}\bar{\mathcal{T}}$ term arrives just in time to guarantee subluminality, and its coefficient cannot vanish but must satisfy, roughly, $\lambda_{\mathcal{T}\bar{\mathcal{T}}} \gtrsim |\lambda_i|^{2/(\Delta_i-2)}$. In the absence of irrelevant scalars of dimension $2 \leq \Delta_i < 4$ and if $\lambda_{\mathcal{T}\bar{\mathcal{T}}}$ is fine-tuned to zero, a similar statement can be made for the coefficient of the $\mathcal{T}\mathcal{T}\bar{\mathcal{T}}\bar{\mathcal{T}}$ term, etc.

The fact that sound approaches the speed of light at low temperatures implies a strong bound on thermalization, as in the high temperature limit discussed in Sec. 4. Depending on whether the theory has an irrelevant scalar $\mathcal{O}_i$ with dimension $2 \leq \Delta_i < 3$, the bound will either be controlled by that scalar deformation or the $\mathcal{T}\bar{\mathcal{T}}$ deformation, and one has

$$\tau_{\text{eq}} \gtrsim \frac{1}{T} \frac{1}{c_{\text{IR}}} \left( \frac{\Lambda}{T} \right)^p , \qquad \text{with} \quad \begin{cases} p = 2(\Delta-2), & \text{if } \Delta \equiv \min_i \Delta_i < 3 , \\ p = 2, & \text{if } \Delta \geq 3 , \end{cases} \tag{57}$$

for $T \ll \Lambda$. Following the arguments made in Sec. 4.1, one can show that this bound is typically saturated if $\min_i \Delta_i < 3$. Instead, if $\mathcal{T}\bar{\mathcal{T}}$ controls the leading correction to the equation of state, the bound is loose: sound is made subluminal without a proportional decrease in the equilibration time.

## 5.3 Extra symmetries

The results obtained so far straightforwardly generalize to QFTs with additional global symmetries, abelian or non-abelian. Each continuous symmetry will lead to a new hydrodynamic degree of freedom; while these have subtle dynamics because of the large hydrodynamic fluctuations [22, 24, 25], the KPZ sound mode is essentially unchanged and our bound (4) follows.

This logic even applies to QFTs coupled to a finite chemical potential $\mu$. We focus here on a $U(1)$ symmetry for concreteness. In higher dimensions, CFTs at finite density can have a non-trivial equation of state $P(T, \mu) = T^d f(\mu/T)$. However in $d = 2$ the Virasoro and current algebra entirely fixes the equation of state (since correlators of spin-1 currents on the cylinder are completely fixed by 2d conformal invariance) to [54–56]

$$P(T, \mu) = \frac{\pi}{6} c T^2 + \frac{k}{2} \mu^2 . \tag{58}$$

The densities can be obtained from the pressure as $dP = sdT + nd\mu$ – Eq. (58) leads to $(\delta s\ \delta n) = \chi\left(\begin{smallmatrix}\delta T\\\delta\mu\end{smallmatrix}\right)$ with a diagonal and constant matrix of susceptibilities

$$\chi_{ss} = \frac{\pi}{3}c\,, \qquad \chi_{nn} = k\,, \qquad \chi_{sn} = 0\,. \tag{59}$$

The speed of sound at finite density is given by (see, e.g., [16])

$$c_s^2 = \left.\frac{dP}{d\varepsilon}\right|_{S,N} = \frac{(s\ n)\,\chi^{-1}\left(\begin{smallmatrix}s\\n\end{smallmatrix}\right)}{sT + n\mu}\,. \tag{60}$$

Evaluating this for a constant susceptibility matrix gives $c_s^2 = 1$. Two-dimensional CFTs at finite temperature and density therefore have no room for hydrodynamics; consequently, thermalization is also strictly constrained in the high and low temperature limits of finite density QFTs, and similar arguments to those of the previous sections lead to the bound (4) at high temperatures and (57) at low temperatures.

## 5.4 Large $c_{\text{UV}}$ theories[13]

In theories with a large number of degrees of freedom $c_{\text{UV}} \gg 1$, the bound (43) becomes weak, but a diffusion bound of the form (45) still gives a strong constraint at high temperatures. Physically, hydrodynamic fluctuations are suppressed by thermodynamic susceptibilities which scale with $c_{\text{UV}}$, so that when $c_{\text{UV}} \gg 1$ the hydrodynamic interactions that led to KPZ can be ignored, and the system is diffusive. We discuss the relevant bound for this situation in more detail in this section, and highlight an important subtlety in applying causality bounds to thermalization.

First note that, strictly, the inequality (3) bounds the time scale suppressing higher derivative corrections near the lightcone, as these are the corrections that need to be large enough to restore causality. Indeed, evaluating a retarded Green's function along the trajectory

$$x(t) = c_s t + (\mathcal{D}t)^{1/z}\,, \tag{61}$$

one expects to find corrections of the form

$$G^R_{T_{0x}T_{0x}}(t, x(t)) \sim \frac{sT}{t(t\mathcal{D})^{1/z}}\left[1 + \left(\frac{\tau_{\text{correction}}}{t}\right)^{\#} + \cdots\right]\,. \tag{62}$$

When $t < \mathcal{D}^{1/(z-1)}/(1-c_s)^{z/(z-1)}$, the operators are spacelike separated and the retarded Green's function must vanish – for this to be possible the correction must be large enough, i.e.,

$$\tau_{\text{correction}} \gtrsim \frac{\mathcal{D}^{1/(z-1)}}{(1-c_s)^{z/(z-1)}}\,. \tag{63}$$

One may expect that corrections are suppressed by the UV cutoff of hydrodynamics, $\tau_{\text{correction}} \sim \tau_{\text{eq}}$, which would then lead to Eq. (3). While this is correct for KPZ hydrodynamics, it is not in diffusive hydrodynamics: we show below that in this case the correction is $\tau_{\text{correction}} \sim \tau_{\text{eq}}^2/t_D$, with the diffusion time defined as[14]

$$t_D \equiv \frac{D}{c_s^2}\,. \tag{64}$$

---

[13]We are thankful to Richard Davison for discussions that led to the identification of a mistake in a previous version of this section. We also thank Hesam Soltanpanahi for related discussions.

[14]A similar subtlety does not arise in KPZ hydrodynamics relevant for 2d QFTs at finite $c_{\text{UV}}$. There, irrelevant corrections to diffusion can be ignored, and KPZ diffusion arises from relevant interactions in the coordinates that follow the pulse $x = \pm c_s t$. The dynamics no longer depends on $c_s$, nor therefore on the timescale $\mathcal{D}^2/c_s^3$ analogous to $t_D$. Irrelevant correction instead arise from corrections to KPZ scaling.

To do this, we study higher derivative corrections to hydrodynamics – these are expected to arise at the scale set by the cutoff and can be used as a proxy for $\tau_{\text{eq}}$. In 1+1d, there is a single term, $\tau_\Pi$, that is second order in derivatives in the constitutive relation [57]:

$$T^{\mu\nu} = \varepsilon u^\mu u^\nu + P\Delta^{\mu\nu} - \zeta(1 - \tau_\Pi u \cdot \partial)(\partial \cdot u)\Delta^{\mu\nu} + \cdots, \tag{65}$$

with $\Delta^{\mu\nu} = \eta^{\mu\nu} + u^\mu u^\nu$. Linearizing the continuity relations $\partial_\mu T^{\mu\nu} = 0$ as in Sec. 3 leads to the following equation of motion for the right-moving mode $\pi_+ = \delta\pi + c_s \delta\epsilon$:

$$(\partial_t + c_s \partial_x)\pi_+ - \frac{1}{2}D(1 - \tau_\Pi \partial_t)\partial_x \partial_t \pi_+ = 0, \tag{66}$$

with $D = \zeta/(sT)$. We have dropped a higher derivative mixing term with the left-moving mode $\pi_-$, as this will give exponentially suppressed contributions in the kinematics (61). Defining

$$\tilde{x} \equiv \frac{x - c_s t}{\sqrt{D}}, \tag{67}$$

the equation becomes

$$\partial_t \pi_+ + \frac{1}{2}\left[1 + \left(\frac{1}{2}\sqrt{t_D} + \frac{\tau_\Pi}{\sqrt{t_D}}\right)\partial_{\tilde{x}}\right]\partial_{\tilde{x}}^2 \pi_+ = 0. \tag{68}$$

These coordinates have diffusive scaling $t \sim \tilde{x}^2$. The equation above reveals the time scales suppressing higher derivative corrections along the trajectory (61): $t_D$ and $\tau_\Pi^2/t_D$, instead of the cutoff $\tau_{\text{eq}} \sim \tau_\Pi$. Along this trajectory the Green's function takes the form (dropping numerical factors)

$$G_{T_{0x}T_{0x}}^R(t, x(t)) \sim \frac{sT}{t(tD)^{1/2}}\left[1 + \left(\frac{t_D + \tau_\Pi^2/t_D}{t}\right)^{1/2} + \cdots\right], \tag{69}$$

as can be verified by direct calculation. The bound (63) therefore applies to the timescale $\tau_{\text{correction}} \sim \max(t_D, \tau_\Pi^2/t_D)$. At high temperatures one approaches the CFT, so that $c_s \to 1$ and $t_D \to 0$ (because the bulk viscosity vanishes in the CFT). The bound then becomes

$$\tau_{\text{eq}} \gtrsim \frac{t_D}{1 - c_s}. \tag{70}$$

Alternatively, this bound can be expressed as an upper bound on the bulk viscosity

$$\frac{\zeta}{s} \lesssim (1 - c_s)T\tau_{\text{eq}}. \tag{71}$$

This bound applies in particular to holographic 2d CFTs deformed by a relevant operator. Identifying $1/\tau_{\text{eq}}$ with the frequency of the first quasinormal mode [2, 58–60] $\omega_{\text{qnm}} \sim T$, one finds that (71) is a strong bound on the diffusion constant (or, more precisely, the sound attenuation rate) at high temperatures $T \gg \Lambda$:

$$\frac{\zeta}{s} \lesssim (1 - c_s) \simeq (2 - \Delta)\alpha_\Delta \left(\frac{\Lambda}{T}\right)^{2(2-\Delta)}. \tag{72}$$

It is interesting to contrast this with a conjectured lower bound on the bulk viscosity [61], which in 2d reads $\frac{\zeta}{s} \geq \frac{1}{\pi}(1 - c_s)$. Although this latter bound is known to be violated in certain situations [62, 63], the bulk viscosity in holographic models typically behaves as $\zeta/s \sim (1 - c_s)$ [62, 64, 65]. From the perspective of Eq. (71), these systems therefore thermalize as rapidly as allowed by causality, in 2d. It would be interesting to further explore asymptotically AdS$_3$ bulks with relevant deformations from this perspective; see Refs. [66–69] for partial results in that direction[15].

---

[15]In particular, it is surprising that the hydrodynamic Green's function (69) receives large corrections even at timescales parametrically larger than the equilibration time $\tau_{\text{eq}} \sim \tau_\Pi$ at which 'new physics', in the form of quasi-normal modes, arises. We leave further investigation of this issue for future work.

# 6 Future Directions

We have found that (1+1)d QFTs thermalize slowly, with equilibration time bounded below. In a similar vein, bounds on transport [70,71], chaos [72] and the equilibration time [2,6,73,74] constrain the thermal dynamics of quantum systems without relying on a quasiparticle picture. One distinguishing feature of our bound (4) is that temperature is not the only scale involved – 2d QFTs therefore have 'sub-Planckian' thermalization at high and low temperatures despite the absence of a quasiparticle description, in contrast to what is expected in higher dimensions or in systems without momentum conservation [6,73].

We have focused on QFTs obtained by perturbing UV CFTs with a relevant operator; another possible way to define a QFT is as the continuum limit of a lattice model. In such theories the spacetime symmetries of the QFT, in particular translation invariance, are only emergent. This will entirely change the high temperature limit of the theory, and momentum will not be a long lived collective excitation in the hydrodynamic regime. However, our results may apply to lattice systems close to a critical point described by a CFT, in particular in the quantum critical fan region, if a sufficiently large parametric window exists between the lattice scale $a$ and the scale of the relevant deformation: $\Lambda \ll T \ll c_s/a$. KPZ dissipation was observed numerically in a classical spin chain at intermediate temperatures in Ref. [75]. Our results also generalize to Lorentz breaking relevant deformations, as long as the equation of state approaches that of a CFT at high (or intermediate) temperatures.

# Acknowledgements

We thank Bruno Balthazar, Daniel Brennan, John Cardy, Anushya Chandran, Clay Córdova, Richard Davison, Christian Ferko, Blaise Goutéraux, Sašo Grozdanov, Edward Mazenc, Anatoli Polkovnikov and Hesam Soltanpanahi for enlightening discussions. We also thank Tom Hartman and Sean Hartnoll for valuable comments on an earlier version of this manuscript. LVD is supported by the Swiss National Science Foundation and the Robert R. McCormick Postdoctoral Fellowship of the Enrico Fermi Institute. ALF and EK were supported in part by the US Department of Energy Office of Science under Award Number DE-SC0015845 and the Simons Collaboration Grant on the Non-Perturbative Bootstrap, and ALF in part by a Sloan Foundation fellowship. MTW is partly supported by the Simons Collaboration on the Nonperturbative Bootstrap and the National Centre of Competence in Research SwissMAP funded by the Swiss National Science Foundation.

# A Special Cases

## A.1 Free fermion and scalar

The equation of state of a two-dimensional free fermion or scalar is given by

$$P = \sigma T \int \frac{dk}{2\pi} \log\left(1 + \sigma e^{-\beta\sqrt{m^2+k^2}}\right), \tag{73}$$

with $\sigma = +1$ for the fermion and $\sigma = -1$ for the boson. For the scalar, one finds that in the high temperature limit $T \gg m$ the pressure is

$$P = \frac{\pi}{6}T^2 - \frac{1}{2}mT + \cdots. \tag{74}$$

We will further comment on this system in Appendix A.3.

For the fermions, the mass term $\delta S = m \int d^2 x \bar{\psi} \psi$ is a relevant deformation with dimension $\Delta = 1$. At high temperatures $T \gg m$ the pressure is given by

$$P = \frac{\pi}{12} T^2 - \frac{m^2}{4\pi} \log \frac{T}{m} + \cdots , \tag{75}$$

leading to

$$s_o = \frac{\pi}{6} \left[ 1 - 6 \frac{(m/2\pi)^2}{T^2} + \cdots \right] , \tag{76}$$

which agrees with the general result (16) from conformal perturbation theory, with $c_{\text{UV}} = \frac{1}{2}$, $\Delta = 1$ and $\lambda = \frac{m}{\sqrt{2}\pi}$ (this last identification follows from the fact that the CFT primary in (12) is normalized as $(\bar{\psi}\psi)_{\text{CFT}} = \sqrt{c_{\text{UV}}} 2\pi \bar{\psi}\psi$). Note that the pressure (75) has a logarithmic enhancement – correspondingly, there is a divergence in the conformal perturbation theory expression (11) for the pressure when $\Delta \to 1$.

## A.2 Integrable Ising flows

The critical Ising model (Ising field theory) deformed by the magnetic $\sigma$ operator,[16]

$$S = S_{\text{IFT}} - \frac{h}{\sqrt{c_{\text{UV}}}} \int_0^\beta dt \int_{-\infty}^\infty dx \, \sigma(x), \tag{77}$$

is integrable [76], providing probably the simplest example of a nontrivial but solvable CFT deformed by a relevant operator. The $\sigma$ operator is a scalar with dimension $\Delta = \frac{1}{8}$ and the UV central charge is $c_{\text{UV}} = \frac{1}{2}$, so according to conformal perturbation theory the pressure at high $T$ is

$$P = \frac{c_{\text{UV}} \pi T^2}{6} + h^2 \frac{4\Gamma\left(\frac{7}{16}\right)\Gamma\left(\frac{17}{16}\right)}{\pi^{3/4} T^{7/4} \Gamma\left(\frac{9}{16}\right)\Gamma\left(\frac{15}{16}\right)} + \mathcal{O}(h^3) = \frac{\pi T^2}{12}\left(1 + 7.71\left(\frac{h}{T^{\frac{15}{8}}}\right)^2 + \mathcal{O}(h^3)\right). \tag{78}$$

In the opposite limit, $T \to 0$, the pressure is dominated by a temperature-independent vacuum energy $\Lambda_{\text{IR}}$, which does not affect the speed of sound $c_s$. The leading temperature-dependent contribution comes from the lightest particle, with mass $m_1$, propagating around the thermal circle:

$$\delta P = -\frac{T^2}{2\pi} \int_{-\infty}^\infty dx \log\left(1 - e^{-\sqrt{x^2 + m_1^2/T^2}}\right) \approx \frac{T^2}{2\pi} \sqrt{\frac{2\pi m_1}{T}} e^{-\frac{m_1}{T}}, \tag{79}$$

and so at small $T$, $c_s^2 \approx \frac{T}{m_1}$. By scaling, the mass $m_1$ is proportional to a power of $h$, and in this case, the proportionality constant can be computed exactly:

$$m_1 = 4.402 h^{8/15}. \tag{80}$$

For intermediate values of $T$ away from these limits, the pressure can be computed using the Thermodynamic Bethe Ansatz, as in [77]; the scaling function $\tilde{c}(r)$ computed in Table 2 of [77] is related to the pressure by $P(T) = \frac{\tilde{c}\left(\frac{m_1}{T}\right)\pi T^2}{6} - \Lambda_{\text{IR}}$. Using this result, in units where $h = 1$, we show the speed of sound and dimensionless entropy $s_o$ as a function of $\beta$ in Fig. 3. As expected, the exact result smoothly interpolates between the two limits, and in fact is well-approximated by one or the other limit for most values of $T$.

---

[16]We are dividing $h$ in the action by $\sqrt{c_{\text{UV}}}$ for simpler comparison with the literature, since this factor compensates for our convention in this paper that $\langle \mathcal{O}(x)\mathcal{O}(0)\rangle \sim \frac{c_{\text{UV}}}{x^{2\Delta}}$.

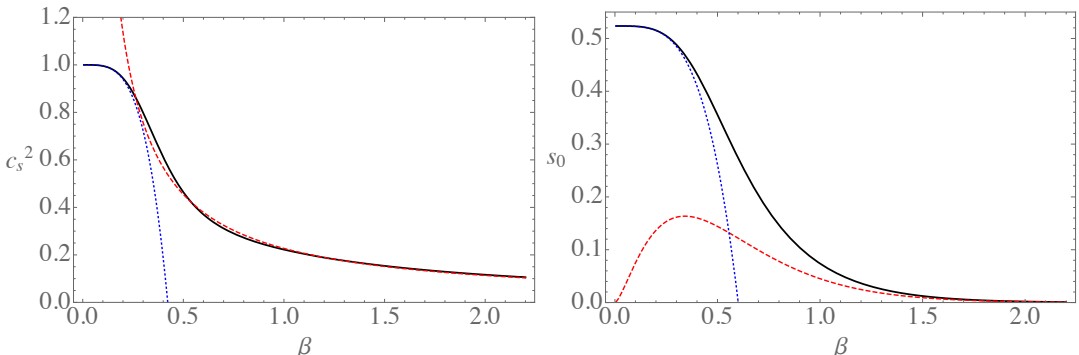

Figure 3: *Left:* Speed of sound squared, $c_s^2 = \frac{\frac{dP}{dT}}{T\frac{d^2P}{dT^2}}$, as a function of inverse temperature $\beta$ in Ising Field Theory deformed by the $\sigma$ operator in units with $h = 1$. The black, solid line is the exact result from integrability. Also shown are the leading large $T$ result in conformal perturbation theory (blue, dotted) from (78) and the leading small $T$ result $c_s^2 \approx \frac{1}{m_1\beta}$ (red, dashed), where $m_1$ is the mass of the lightest particle. *Right:* Dimensionless entropy $s_o = \frac{1}{T}\frac{dP}{dT}$ as a function of $\beta$. Again, the black solid line is the exact result in the Ising Field Theory deformed by $\sigma$ with $h = 1$, shown together with the leading conformal perturbation theory result (blue, dotted), and leading small $T$ result $s_o \approx \frac{e^{-\beta m_1}(\beta m_1)^{3/2}}{\sqrt{2\pi}}$ (red, dashed).

Another integrable flow with the Ising model at an endpoint is the flow from the Tricritical Ising Model (TIM) to Ising:

$$S = S_{\text{TIM}} + \frac{g}{\sqrt{c_{\text{UV}}}} \int_0^\beta dt \int_{-\infty}^\infty dx\, \epsilon'. \tag{81}$$

The $\epsilon'$ operator has dimension $\Delta = \frac{6}{5}$ and the UV central charge $c_{\text{UV}}$ in this case is the TIM central charge $c_{\text{UV}} = \frac{7}{10}$. The pressure as a function of temperature was computed numerically in [78]. For $g = 1$, we show the speed of sound $c_s$ and dimensionless entropy $s_o$ as a function of $T$ for this flow in Fig. 4. The leading irrelevant operator as one approaches the Ising model in this case is the operator $\mathcal{T}\bar{\mathcal{T}}$, so this is a case where the sign of $\lambda_{\mathcal{T}\bar{\mathcal{T}}}$ is fixed by causality.

## A.3 Scalar + $\phi^p$

Consider the QFT obtained from the free scalar theory with the following relevant deformation

$$S = \int d^2x\, \frac{1}{2}(\partial\phi)^2 + \frac{\lambda}{p!}\phi^p, \tag{82}$$

with $p \geq 2$ even. Although $\phi^p$ is not a primary or descendant of the CFT, its logarithmic two-point function implies that it formally has dimension $\Delta = 0$. The conformal perturbation theory approach used in Sec. 2 predicts corrections to the equation of state that are analytic in coupling $\delta P \propto \lambda^2$ – however the results obtained there diverge when $\Delta \to 0$. In fact, the pressure of free massive scalar (74) already shows that for $p = 2$ the correction is not analytic in coupling $\delta P \propto \sqrt{m^2}$. We will show more generally that the high temperature equation of state of the theory (82) is non-analytic in coupling. This breakdown of conformal perturbation theory stems from the absence of a thermal mass of the two-dimensional massless scalar, which leads to IR divergences in the expansion (11).

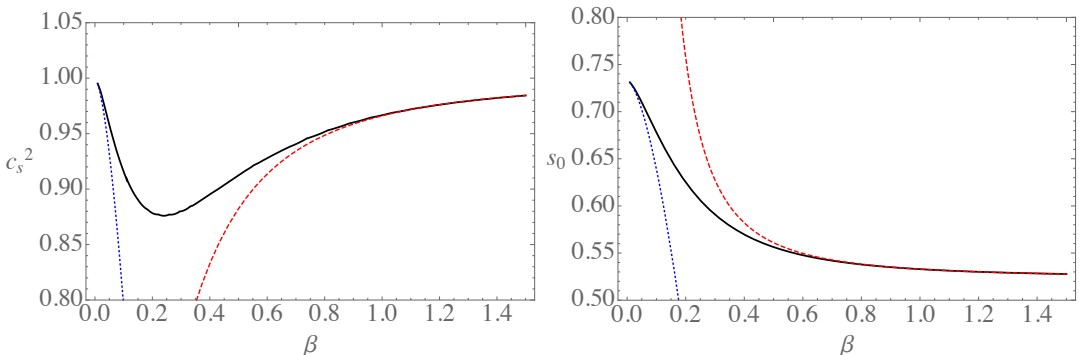

Figure 4: Same as Fig. 3, but for flow from tricritical Ising model deformed by $\epsilon'$ to the critical Ising model. Red, dashed lines in this case are the leading correction in the IR due to the $\mathcal{T}\bar{\mathcal{T}}$ deformation around Ising; blue, dotted lines are leading correction in UV from conformal perturbation theory; and black, solid line is numeric result from [78].

We start with $p = 4$ and generalize later. When computing the partition function on the thermal cylinder, it is convenient to expand the scalar field into a zero mode and KK modes

$$\phi(x, \tau) = \phi_0(x) + \sum_{n \neq 0} e^{i\omega_n \tau} \phi_n(x), \tag{83}$$

with Matsubara frequencies $\omega_n = 2\pi n T$, see e.g. Refs. [45,79]. The dimensionally reduced action reads, after canonical normalization

$$S = \int d^{d-1}x \, \frac{1}{2}(\nabla\phi_0)^2 + \frac{\lambda T}{4!}\phi_0^4 + \sum_{n>0}\left(|\nabla\phi_n|^2 + \omega_n^2|\phi_n|^2 + \frac{\lambda T}{2}\phi_0^2|\phi_n|^2 + \cdots\right). \tag{84}$$

There are also two other interaction terms: $\lambda T \phi_0 \phi_n^3$ and $\lambda T \phi_n^4$. We have generalized to arbitrary spacetime dimension, and will restore $d = 2$ below. The KK modes have mass $\omega_n$ and interaction $\lambda T$. Their dimensionless interaction is therefore

$$\frac{\lambda T}{\omega_n^{5-d}} \sim \left(\frac{\Lambda}{T}\right)^{4-d}, \tag{85}$$

with $\Lambda \equiv \lambda^{1/(4-d)}$. In $d < 4$, where the interaction is relevant, the KK modes are weakly coupled at high temperatures $T \gg \Lambda$. Let us now turn to the zero mode. It acquires a thermal mass from its coupling to the KK mode shown in (84) – to leading order in $\lambda$ one finds, after performing the Matsubara sum

$$m_{\text{th}}^2 = \frac{\lambda T}{2} \int \frac{d^{d-1}k}{(2\pi)^{d-1}} \frac{\beta}{2|k|} \coth\frac{\beta|k|}{2} - \frac{1}{k^2}. \tag{86}$$

This integral is free of IR divergences since only KK modes are running in the loop, and temperature-independent UV divergences can be removed with a mass counterm in (82). In $d = 4$, this integral leads to the well known thermal mass $m_{\text{th}}^2 = \frac{1}{4!}\lambda T^2$ – this implies that the zero-mode dimensionless coupling $\lambda T/m_{\text{th}}^{5-d}$ is also small at high temperatures (if $\lambda$ is small). In $d = 3$, the thermal mass is $m_{\text{th}}^2 = -\frac{\lambda T}{4\pi}\log\frac{\Lambda}{T}$ [79], implying that the zero-mode is only logarithmically weakly coupled at high temperature. Finally, in the case $d = 2$ of interest one finds a thermal mass

$$m_{\text{th}}^2 = \frac{\lambda}{4\pi}\log\frac{\Lambda}{T}. \tag{87}$$

The zero-mode is therefore *strongly* coupled at high temperatures $T \gg \sqrt{\lambda}$. This explains the issues encountered in conformal perturbation theory (11) (which is a weak coupling expansion) when $\Delta = 0$. Similar infrared effects lead to a breakdown of perturbation theory in high temperature QCD [36]; they are stronger for the two-dimensional theory (82) and already dominate the leading correction to the high-temperature equation of state.

Let us study the contribution of the strongly coupled zero-mode to the pressure. After integrating out the KK modes, the partition function takes the form

$$Z = e^{\beta V P_{KK}} \int D\phi_0 \, e^{-\int dx \frac{1}{2}\phi_0(m_{\text{th}}^2 - \nabla^2)\phi_0 + \frac{\lambda T}{4!}\phi_0^4}, \tag{88}$$

so that the pressure $P = \frac{T}{V} \log Z$ is

$$\begin{aligned} P &= P_{\text{KK}} + P_{\text{zm}} \\ &= \frac{\pi}{6} T^2 \left(1 + O(\lambda/T^2)\right) - T E_{\text{gs}}(m_{\text{th}}^2, \lambda T), \end{aligned} \tag{89}$$

where $E_{\text{gs}}(m^2, g)$ is ground state energy of the $(0+1)$-dimensional anharmonic oscillator $\frac{1}{2}\dot{\phi}^2 + \frac{1}{2}m^2\phi^2 + \frac{1}{4!}g\phi^4$ (we note in passing that setting $m_{\text{th}} \to m$ and $\lambda \to 0$ reproduces the equation of state of the free scalar (74), since $E_{\text{gs}}(m^2, 0) = \frac{1}{2}m$). At high temperatures, $E_{\text{gs}}$ will have a strong coupling expansion which by dimensional analysis must take the form

$$E_{\text{gs}}(m_{\text{th}}^2, \lambda T) = (\lambda T/4!)^{1/3} \left[ a_0 + a_1 \left( \frac{m_{\text{th}}^2}{(\lambda T/4!)^{2/3}} \right) + a_2 \left( \frac{m_{\text{th}}^2}{(\lambda T/4!)^{2/3}} \right)^2 + \cdots \right]. \tag{90}$$

Although the zero-mode sector is strongly coupled, as a quantum mechanical system it is well amenable to numerics; see e.g. Ref. [80], which found $a_0 \approx 0.667986$ and $a_1 \approx 0.143668$. We therefore find that the leading correction to the equation of state at high temperature is

$$P = \frac{\pi}{6} T^2 - a_0 T (\lambda T/4!)^{1/3} + \cdots. \tag{91}$$

Notice that it has the right sign required by subluminality of sound, as found in Sec. 2. Unlike the other special cases studied in appendices A.1 and A.2 which are integrable, $\phi^4$ theory is non-integrable and expected to have emergent KPZ hydrodynamics at finite temperature and late time. Following similar arguments to those in the main text, this leads to a strong bound on the equilibration time of this theory at high temperatures $T \gg \Lambda \equiv \sqrt{\lambda}$

$$\tau_{\text{eq}} \gtrsim \frac{1}{T} \left( \frac{T}{\Lambda} \right)^{2/3}. \tag{92}$$

This result can be generalized to the theory (82) with other values of the exponent $\phi^p$, fine-tuning lower powers away. By dimensional analysis, the ground state of the quantum mechanical system $H = \frac{1}{2}\dot{\phi}^2 + \lambda T^{\frac{p-2}{2}}\phi^p$ is $E_{\text{gs}} \sim (\lambda T^{\frac{p-2}{2}})^{\frac{2}{p+2}}$ leading to a correction to the pressure

$$P = \frac{\pi}{6} T^2 - a_0' T (\lambda T^{\frac{p-2}{2}})^{\frac{2}{p+2}}. \tag{93}$$

The equilibration time of this QFT is therefore bounded by

$$\tau_{\text{eq}} \gtrsim \frac{1}{T} \left( \frac{T}{\Lambda} \right)^{\frac{4}{p+2}}. \tag{94}$$

# B  Comparison with Free Energy as a $C$-theorem

In this paper, we have discussed the bound on the speed of sound $c_s^2 \leq 1$ as a $C$-theorem, since it implies the dimensionless entropy density $s_o \equiv \frac{s}{T} = \frac{1}{T}\frac{dP}{dT}$ is monotonic and equal to $\frac{\pi c}{3}$ at fixed points. This is closely related to, but different from, a similar $C$-theorem in the literature from considering the dimensionless free energy [8–12]:

$$f(T) \equiv \frac{P(T)}{T^2}. \tag{95}$$

In defining the above quantity, it is important that zero-temperature vacuum energy density (aka cosmological constant) $\Lambda_{\text{IR}}$ be set to zero. Setting it to zero is equivalent to tuning the constant term $\Lambda_{\text{IR}}$ in the Lagrangian at low energies to zero. Since $\Lambda$ usually preserves all symmetries of the theory (aside from supersymmetries), it is generated along RG flows and therefore setting it to zero in the IR is not the same as setting it to zero in the UV. Therefore, to compute $f(T)$ in practice in a CFT with a relevant deformation, we must solve the theory deep in the IR where the relevant deformation is strongly coupled, *even if we only want to know $f(T)$ at high temperatures where the relevant deformation is weakly coupled*.

To see why the value of $\Lambda$ affects the monotonicity of $f(T)$, note that

$$T^3 \frac{df}{dT} = \varepsilon(T) - P(T), \tag{96}$$

where we have used the relation $T\frac{dP}{dT} = \varepsilon + P$. Therefore, $f(T)$ is monotonic if and only if $\varepsilon \geq P$. However, $\Lambda$ contributes to $P$ and $\varepsilon$ with opposite sign, $P(T) = -\varepsilon(T) = -\Lambda$. So by shifting the value of $\Lambda$, we can make $\varepsilon - P$ take any value we want. In a sense, $\frac{df}{dT} > 0$ is a scheme-dependent statement, since it is true or false depending on our prescription for the bare value of $\Lambda_{\text{UV}}$ in the UV.

To emphasize this point, consider the value of $\Lambda$ in CFTs deformed by strongly relevant $(0 < \Delta < 1)$ operators. In this case, there are no UV divergences in the theory and a natural choice is to set the bare value $\Lambda_{\text{UV}}$ in the UV Lagrangian to zero. However, from equation (14), we see that the leading correction to $f(T)$ in conformal perturbation theory gives

$$T^3 \frac{df}{dT} = -\frac{\pi c_{\text{UV}} \alpha_\Delta (2-\Delta)}{3(1-\Delta)} \left(\frac{\lambda}{T^{1-\Delta}}\right)^2 + 2\Lambda_{\text{UV}} + \dots, \tag{97}$$

where $\dots$ are subleading at large $T$. Therefore, if $0 < \Delta < 1$ and $\Lambda_{\text{UV}} = 0$, then $\frac{df}{dT} < 0$ at sufficiently large $T$, violating its claimed monotonicity.

Take for instance the Ising model with a magnetic field deformation (77) as an explicit example. The deformation $\sigma$ in this case has $\Delta = 1/8$, so $f(T)$ will not be monotonic if we set $\Lambda_{\text{UV}} = 0$. Because the RG flow is integrable, the IR value $\Lambda_{\text{IR}}$ is known exactly:

$$\Lambda_{\text{IR}} - \Lambda_{\text{UV}} = -1.1976 h^{16/15}. \tag{98}$$

Usually in the integrability literature, $\Lambda_{\text{UV}}$ is implicitly set to vanish, and one finds $\Lambda_{\text{IR}}$ in that case is nonzero and negative. However, in the definition of $f(T)$ we are suppose to set $\Lambda_{\text{IR}} = 0$, so we must set $\Lambda_{\text{UV}} = 1.1976 h^{16/15} > 0$, which pushes $\frac{df}{dT}$ in (97) to be positive again.

In contrast, the $C$-function we consider in this paper, $s_o = \frac{1}{T}\frac{dP}{dT}$, does not depend on the value of $\Lambda$ since $\Lambda$ contributes a constant to $P$. Therefore, it can be computed (or measured!) at a temperature $T$ using only the value of observable quantities at that temperature. Moreover, monotonicity of $s_o$ implies monotonicity of $f(T)$ (with $\Lambda$ chosen as described above), as follows. From (17), monotonicity of $s_o$ is equivalent to $c_s^2 < 1$, which is equivalent to $1 > c_s^2 = \frac{dP}{d\varepsilon} = \frac{dP/dT}{d\varepsilon/dT}$. Therefore, monotonicity of $s_o$ implies that $\varepsilon$ increases faster as a function of $T$ than $P$ does, $\frac{d}{dT}(\varepsilon - P) \geq 0$. If $\Lambda$ is tuned to zero in the IR, so that $\varepsilon = P$ at $T = 0$, then it follows that $\varepsilon \geq P$ at all $T$, which by (96) implies $\frac{df}{dT} \geq 0$.

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
