# Peer review of "Thermalization and Hydrodynamics of Two-Dimensional Quantum Field Theories"

_SciPost Physics, doi:SciPost Phys. 12, 119 (2022)_

## Round 1 · Referee Report · Anonymous (Referee 1) · 2021-7-1

Report
This paper adds to the list of recent investigations regarding bounds on characteristics of thermal quantum many-body systems based on fundamental QFT principles. The authors investigate the onset time after which only excitations of conserved quantities are important in relevant deformations of thermal CFTs. The lower bound for this hydrodynamic "equilibration time" puts a fundamental constraint on thermalization. It depends on the diffusion constant as it relies on balancing causality of the QFT against superluminality of diffusive hydrodynamics at very early times.
At a technical level, the analysis relies on conformal perturbation theory around the UV CFT (e.g. to obtain a correction to the conformal equation of state). Several extensions (such as higher dimensions, additional symmetries etc.) as well as specific examples (such as free fields and integrable flows) are discussed.
The content is original, should be of interest to the community, and seems to be correct. The paper is comprehensive, very well written, and nicely structured. Figures are clear and helpful. I recommend publication basically as is. My only suggestion would be to make some parts of the manuscript a bit more easily accessible. As it is written, the content is sometimes quite dense and perhaps not always easy to follow for a more "casual" reader. To give two specific examples: I found the reasoning around Eqs. (8), (9) and (47) a bit hard to digest unless one is already intimately familiar with it. I will consider this suggestion as optional for the authors to take into account, as it will merely improve readability for a certain audience, but is not strictly necessary from a scientific point of view.
At a technical level, the analysis relies on conformal perturbation theory around the UV CFT (e.g. to obtain a correction to the conformal equation of state). Several extensions (such as higher dimensions, additional symmetries etc.) as well as specific examples (such as free fields and integrable flows) are discussed.
The content is original, should be of interest to the community, and seems to be correct. The paper is comprehensive, very well written, and nicely structured. Figures are clear and helpful. I recommend publication basically as is. My only suggestion would be to make some parts of the manuscript a bit more easily accessible. As it is written, the content is sometimes quite dense and perhaps not always easy to follow for a more "casual" reader. To give two specific examples: I found the reasoning around Eqs. (8), (9) and (47) a bit hard to digest unless one is already intimately familiar with it. I will consider this suggestion as optional for the authors to take into account, as it will merely improve readability for a certain audience, but is not strictly necessary from a scientific point of view.

Author: Luca Delacrétaz on 2021-09-07 [id 1741]
(in reply to Report 1 on 2021-07-01)We thank the referee for the positive feedback. Following their suggestions, we have added several comments in the v2 to guide the more casual reader.
In the interest of the reader, we would like to clarify that our bound does not depend on the diffusion constant -- the equilibration time is lower bounded by entirely thermodynamic quantities. This arises because of the large hydrodynamic fluctuations in (1+1)d.

---

## Round 2 · Referee Report · Anonymous · 2021-12-29

Report
The paper examines hydrodynamic behaviour in two dimensional QFTs, viewed as CFTs deformed by a relevant operator. The single hydrodynamic mode in the system is the propagating sound mode, arising from the broken conformal symmetry. The aim is to translate a bound derive for the onset of equilibration times in diffusive systems to this case, where the long-time behavior of the sound mode is modified by strong IR effects. The authors translate the known results into a bound for the equilbriation time.
Requested changes
Overall I think the paper is reasonably written and the results ought to be interesting to the community. I think it deserves publication.
That said, I have a couple of suggestions for the authors to consider:
1. By definition, hydrodynamics is a description that is only valid at low energies compared to temperature, so a qualitative statement that says $\tau_{eq} \grsim T^{-1}$ does not have much content. Early time physics is dominated by transient effects and this may be worth noting.
2. In section 3 the authors argue that in higher dimensions late time physics is governed by diffusion and that the hydrodynamic modes are irrelevant. This is not strictly speaking accurate owing to again to IR effects, known in hydrodynamics as long-time tails, cf., https://arxiv.org/abs/1205.5040
3. It would be clearer if the discussion in Section 4.1 began by noting that that the UV physics is governed by the CFT the authors start with with further specialization subsequently to the free case.
4. Hydrodynamics in holographic theories in 2 dimensions has been discussed in https://arxiv.org/abs/1008.4350, which would worth commenting on in Section 5.4
Author: Luca Delacrétaz on 2022-02-02 [id 2142]
(in reply to Report 1 on 2021-12-29)We thank the second referee for the useful feedback. Following their suggestions, we have made clarifications in Secs. 3 and 4.1. We also thank the referee for bringing arxiv:1008.4350 to our attention, which should indeed have been cited in Sec. 5.4; this is now remedied. Finally, we address the two remaining comments:
The motivation for studying or bounding the equilibration time is precisely to understand when the "early time physics [...] dominated by transient effects" gives way to hydrodynamics. Hydrodynamics, like any EFT, is indeed valid only for energies up to some cutoff, but it is not obvious what this cutoff is in a given system. Finding universal relations or bounds for this cutoff has therefore been the subject of active research (see e.g. https://arxiv.org/abs/2107.07802 for a recent review).
Hydrodynamic interactions in d>2, leading to the well-known hydrodynamic long-time tails, are irrelevant in the RG sense (but interesting!). We have clarified in Sec. 3 that we meant "irrelevant" in the technical, and not colloquial, sense.

---

## Round 2 · List of Changes

- Added clarfiying comments above Eqs. (8) and (9).
- Added Footnote 8 explaining the validity of the KPZ correlation function.
- Corrected a statement about the bulk viscosity, Eq. (38).

---

## Round 3 · List of Changes

List of changes: - Added clarifications in Sec. 3 and 4.1. - made corrections to Sec. 5.4 on the application to theories with large c_UV, and added references.

---

## Editorial Decision

published